# Effect of *Malvaviscus arboreus* Flower and Leaf Extract on the Functional, Antioxidant, Rheological, Textural, and Sensory Properties of Goat Yogurt

**DOI:** 10.3390/foods13233942

**Published:** 2024-12-06

**Authors:** Edson Pontes, Vanessa Viera, Gezaildo Silva, Manoel da Silva Neto, Bianca Mendes, Anna Tome, Renata Almeida, Newton C. Santos, Rennan de Gusmão, Hugo Lisboa, Thaisa Gusmão

**Affiliations:** 1Department of Food Engineering, Federal University of Campina Grande, Campina Grande 58429-900, Brazil; edsondspontes@gmail.com (E.P.); biancabsm96@gmail.com (B.M.); annaemanuelle25@gmail.com (A.T.); renata.duarte@tecnico.ufcg.edu.br (R.A.); newtonquimicoindustrial@gmail.com (N.C.S.); rennan.pereira@professor.ufcg.edu.br (R.d.G.); hugom.lisboa80@gmail.com (H.L.); 2Laboratory of Bromatology, Education and Health Centre, Federal University of Campina Grande, Cuité 58175-000, Brazil; vanessa.bordinviera@gmail.com (V.V.); gilsantosnf@hotmail.com (G.S.); 3Department of Biology, Federal University of Rio Grande do Norte, Natal 59078-970, Brazil; pedro.s.neto@outlook.com

**Keywords:** goat dairy, natural dye, unconventional food plants, viscosity, storage

## Abstract

The present study aimed to evaluate the effects of incorporating different concentrations (1% and 2%) of Malvaviscus arboreus flower (FE) and leaf (LE) extracts as functional ingredients in goat milk yogurt. This study analyzed the impact of these formulations (YFE1%, YFE2%, YLE1%, and YLE2%) on the physicochemical, bioactive, antioxidant, rheological, textural, and sensory properties of goat yogurt over a 28-day storage period. Including FE and LE extracts significantly enhanced the yogurt’s antioxidant activity, reaching up to 10.17 µmol TEAC/g, and strengthened its ability to inhibit lipid oxidation during storage. This study also observed a reduction in the viability of lactic acid bacteria, particularly *L. delbrueckii* subsp. *bulgaricus*, suggesting that the extracts may have antimicrobial properties. Notably, using FE, especially at a concentration of 2% (YFE2%), improved both antioxidant and textural properties while reducing syneresis by the end of the storage period. Sensory evaluations showed positive results for YFE1% and YFE2% formulations. These findings suggest that FE has significant potential as a functional food ingredient. This research lays the groundwork for future studies exploring the integration of *Malvaviscus arboreus*-based ingredients into functional food products, opening new possibilities for innovation in this field.

## 1. Introduction

The growing interest in healthy diets has driven the development of new foods that provide essential nutritional value and functional health benefits. In this context, fermented dairy products, particularly those made from goat’s milk, have gained significant traction due to their superior nutritional and functional qualities [1,2,3]. As highlighted by Dantas et al. [4], goat’s milk is a promising alternative to cow’s milk because of its higher concentrations of conjugated linoleic acid, medium-chain fatty acids, and B complex vitamins, such as riboflavin, thiamine, and B12 [5]. Additionally, goat’s milk is well-regarded for its improved digestibility and ability to enhance the absorption of essential minerals, including iron and copper, further solidifying its relevance in dairy production [6].

Among dairy products, yogurt is notable for its wide acceptance, therapeutic benefits, and nutritional qualities, and it is rich in high-quality proteins and essential minerals [7]. To follow the trend toward functional foods, the use of natural extracts obtained from fruits and plants, such as the malvaviscus extract (*Malvaviscus arboreus* Cav.), is an interesting strategy in dairy production, particularly in goat-derived products, as it enhances the nutritional value and bioactive compounds of the food, potentially bringing health benefits to consumers and improving sensory characteristics [4,8,9].

Malvaviscus (*Malvaviscus arboreus* Cav.), an underutilized food plant in human diets, exhibits remarkable nutritional and medicinal attributes [10]. Studies demonstrate its antioxidant capacity, potential hepatoprotective [11], anti-hypertensive effects [12], and gastroprotective action [13]. Additionally, malvaviscus extracts have shown potent activity against microorganisms and parasites, along with notable anti-protease and antioxidant properties [14]. Considering its potential physiological effects, these findings underscore the importance of incorporating malvaviscus into the human diet.

Although Malvaviscus arboreus has been widely recognized for its medicinal and therapeutic properties, such as antioxidant, antimicrobial, and hepatoprotective effects, its direct application in food products remains underexplored [15]. While traditional culinary uses, such as incorporating flowers and fruits into teas, salads, and jams, highlight its potential for natural, health-focused foods, there is limited research on its development into modern food products [16]. This gap is particularly evident in the case of goat yogurt, in which the effects of malvaviscus extracts on quality characteristics, probiotic viability, and bioactivities are still not fully elucidated. Incorporating this extract into goat yogurt could represent a low-cost and effective strategy to enhance shelf life and meet the growing consumer demand for functional foods with added health benefits, thanks to its phytochemical richness and high antioxidant content.

Considering this context, the need to generate data regarding using malvaviscus extracts in developing new dairy products and understanding their effects is highlighted. Hence, this study aims to assess how the malvaviscus flower and leaf extract influence various properties of goat yogurt during refrigerated storage, including its antioxidative, physicochemical, technological, textural, rheological, and sensory characteristics. We anticipate that understanding these effects could open up opportunities to create healthier and more appealing dairy products for consumers who prioritize a balanced, functional diet.

## 2. Materials and Methods

### 2.1. Materials

Pasteurized goat’s milk from the Toggenburg breed was sourced from a cooperative in Barra de Santa Rosa, Paraíba, Brazil. Sugar (Alegre^®^, Usina Monte Alegre S/A, Mamanguape, Brazil) was purchased locally. At the same time, the *Malvaviscus* leaf and flower were harvested at the Federal University of Campina Grande (UFCG/CES), Cuité, Paraíba, Brazil (coordinates: 6°28′53.94″ S and 36°08′58.87″ W). The starter microorganisms (Y472E), which consisted of *Streptococcus salivarius* subsp. *thermophilus* and *Lactobacillus delbrueckii* subsp. *bulgaricus*, along with the probiotic culture *Lacticaseibacillus casei* (BGP93), were commercially obtained from Sacco^®^ company (Campinas, Brazil).

### 2.2. Obtaining Extracts

The *Malvaviscus* specimen was deposited and identified (accession no. 2386) at the UFCG/CES herbarium. Leaves and flowers were collected separately, washed with running water, immersed in a chlorine solution (200 ppm) for 15 min, and rinsed with distilled water. The cleaned materials were then dried in a forced-air circulation oven (Biopar, model S480 AD, Porto Alegre, Brazil) at 60 °C for 24 h before being ground using a Chrome mini-processor (Oster, model 3320, Balneário Piçarras, Brazil).

The flower extract (FE) and leaf extract (LE) were prepared using agitation extraction, following the methodology described by Pontes et al. [17]. The extraction was carried out with 60% cereal alcohol in a 1:10 (*g*/*v*) ratio. The mixture was heated on a hotplate (Fisatom, model 502, São Paulo, Brazil) with continuous stirring using a magnetic stir bar for 60 min at 40 °C. The extract was filtered through 15 mm filter paper (Whatman^®^, GE Healthcare, Chicago, IL, USA) and centrifuged (Novatecnica^®^, model NT 810, Piracicaba, Brazil) at 4000× *g* for 10 min. The resulting supernatant was concentrated using a rotary evaporator (TE-211, Tecnal, Piracicaba, Brazil) at 160 °C, transferred to amber bottles, and stored in a freezer at −18 °C until further analysis [17].

### 2.3. Yogurt Preparation

The yogurt formulations were prepared according to the methodology proposed by Dantas et al. [4]. Five distinct formulations were developed, incorporating different concentrations of flower extract (FE: 1% and 2%) and leaf extract (LE: 1% and 2%) in independent batch replicates. The extracts were added in liquid form to ensure uniform incorporation and distribution within the yogurt mixture. Pasteurized goat milk (initial properties: pH 6.46; carbohydrates 4 g/100 g; protein 3 g/100 g; total fats 3.5 g/100 g; energy value 60 kcal/100 g) was heated to 45 °C. Sugar was added at 100 g/L, followed by inoculation with starter cultures at a concentration of 0.4 g/L and probiotic cultures at 0.1 g/L. The yogurt mixtures were incubated in a biochemical oxygen demand (BOD) incubator at 45 ± 0.5 °C for six hours. The fermentation process was deemed complete when the pH of the yogurts reached 4.5. After fermentation, the yogurts were cooled to 4 ± 0.5 °C for 12 h, following the procedure described by Morais et al. [18]. The cooled yogurts were packaged in sterile containers, with the extracts added as specified for each treatment. The final products were stored in a BOD chamber at 4 ± 0.5 °C and analyzed at three intervals: day 1, day 14, and day 28 of storage.

#### Identification of Formulations

The produced yogurts were coded as follows: control goat yogurt (YC); goat yogurt with 1% addition of LE (YLE1%); goat yogurt with 2% addition of LE (YLE2%); goat yogurt added with 1% of FE (YFE1%); and goat yogurt with 2% addition of FE (YFE2%).

### 2.4. Physical and Physicochemical Parameters Assessed in the Formulations

The water activity was determined through direct reading using an Aqualab meter (Meter^®^, AquaLab Series 4TEV, São José dos Campos, Brazil); moisture content was determined by drying in an oven stabilized at 105 °C until reaching a constant mass; ash content was determined using carbonization, followed by incineration in a muffle furnace (Jung^®^, model 0612, Blumenau, Brazil) stabilized at 550 °C; proteins were determined using the micro-Kjeldahl method; and the acidity in the lactic acid was determined using titration, lactose content, lipids, and pH (Quimis model Q400as) according to the methodology described by AOAC [19]. The carbohydrate content was determined using the difference, as described by Fang and Guo [20]. The energy value was calculated using specific coefficients for the protein, lipid, and carbohydrate fractions [21].

Instrumental color analysis was performed using a colorimeter (Konica Minolta—model CR 400), and the results were expressed as L* (0: dark; 100: white), a* (−a* green; +a* red), and b* (−b* blue; +b* yellow). For each treatment, the mean value was obtained from five readings at different points of three portions (replicates).

### 2.5. Extraction of Bioactive Compounds from Prepared Yogurts

A hydroalcoholic extract was prepared for quantifying bioactive compounds, as described by Dantas et al. [4]. For this, 5 g of each formulation and each of the extracts were added to 15 mL of 80% methanol (Sigma-Aldrich, St. Louis, MO, USA), homogenized for 10 min using a mini-Turrax device (Tecnal^®^), kept at rest for 24 h, filtered with 125 mm filter paper (Whatman^®^, GE Healthcare, Chicago, IL, USA), and analyzed following the methodologies described below.

#### 2.5.1. Total Phenolic and Total Flavonoid Content

The total phenolic content was measured using the Folin–Ciocalteu method [22], and the absorbance was measured at 765 nm with a spectrophotometer (BEL Photonics, Piracicaba, São Paulo, Brazil). The phenolic content was determined using a standard curve prepared with gallic acid (Sigma-Aldrich, St. Louis, MO, USA). The results were expressed in mg of gallic acid equivalents (GAE) per 100 g of sample (mg GAE/100 g).

The total flavonoid content was measured using the procedure described by Zhishen et al. [23]. The sample’s absorbance was measured at 510 nm with a spectrophotometer (BEL Photonics, Piracicaba, Brazil) against a blank (without extract). The total flavonoid content was determined using equivalents from the standard curve of catechin (Sigma-Aldrich). The results were expressed in mg of catechin equivalents (CE) per 100 g of sample (mg CE/100 g).

#### 2.5.2. Antioxidant Activity

The ferric-reducing ability of plasma (FRAP) assay was performed as described by Benzie and Strain [24]. A blank without sample addition was conducted for each treatment. The absorbance was measured at 593 nm using a spectrophotometer (Bel Photonics). A standard curve was generated with 1 mM Trolox, and the results were expressed in micromoles of Trolox equivalent antioxidant capacity (TEAC) per gram (μmol TEAC/100 g).

Antioxidant activity using the ABTS^•+^ (2.2-azino-bis(3-ethylbenzothiazoline)-6-sulfonic acid) method followed the protocol described by Sariburun et al. [25]. Sample absorbance was read at 734 nm after a 6 min reaction. The “blank” solution was the sample’s extraction solvent, which was used to zero the spectrophotometer. Trolox was used as a reference, and the results were expressed in µmol of Trolox equivalent antioxidant capacity (TEAC) per gram of the sample (µmol TEAC/g).

### 2.6. Lipid Oxidation (TBARs)

Lipid oxidation levels were measured using the thiobarbituric acid reactive substances (TBARS) test according to Raharjo et al. [26]. Malondialdehyde (MDA) was extracted from the sample, and readings were taken at a wavelength of 531 nm using a spectrophotometer (BEL Photonics, Piracicaba, São Paulo, Brazil). A control without samples was prepared to zero the spectrophotometer. Results were compared against the blank and expressed in mg of MDA/kg of the sample.

### 2.7. Hygienic–Sanitary Quality and Viability of Lactic Bacteria

Microbiological analyses were performed based on Brazilian legislation specifications [27]. An *Escherichia coli* count, total mold and yeast count, and absence detection of *Salmonella* spp. per 25 g were conducted following Apha [28] guidelines. The viability of lactic bacteria included counts of *S. thermophilus* [28], *L. delbrueckii* subsp. *bulgaricus* [29], and *L. casei* [30]. The results were expressed in colony-forming units per gram (CFU/g).

### 2.8. Texture Profile and Syneresis

All yogurt formulations underwent a texture analysis using a universal texture analyzer (TA-XT plus—Texture Analyzer, Stable Micro Systems equipped with Exponent Stable Micro Systems software 6.2.2.0, using probe P-36R) under the following conditions: pre-test, test, and post-test speeds of 2.0 mm/s, 5.0 mm/s, and 5.0 mm/s, respectively, with a 30 mm distance and a 5 s interval between compressions.

The yogurt’s susceptibility to water separation from the curd was determined using the drainage method described by Riener et al. [31].

### 2.9. Rheological Behavior

The rheological behavior of all formulations was assessed at temperatures of 4, 8, and 12 °C following the experimental procedure of Mitschka [32]. Briefly, 600 mL of each sample were used for readings on a Brookfield viscometer (model DV—II + Pro Brookfield Engineering Laboratories, Middleboro, MA, USA) using a spindle number 2 at rotational speeds of 40, 50, 60, 70, 80, 90, 100, and 120 rpm at temperatures of 4, 6, 8, and 12 °C. The temperature was maintained using a coupled thermostatic bath. Shear stress and strain rates were calculated from the data for different rotational speeds and temperatures. The values were correlated to the Herschel–Bulkley rheological model (Equation (1)).
(1)$$\tau -\tau_{OH}=K_{H}\gamma^{nH}$$
where τ is the shear stress (Pa); $\tau_{OH}$ is the yield stress (Pa); *K_H_* is the consistency index (Pa.s^n^); *_n_H* is the flow behavior index (dimensionless); and γ is the rate of deformation (s^−1^).

### 2.10. Sensory Analysis

At Time 1 of storage, 80 semi-trained panelists (26 men and 57 women, aged between 18 and 33 years, mean 21 ± 2.58 years) were recruited based on their habit of consuming goat yogurt. Potable water and saltine crackers were provided for palate cleansing between sample tastings. For the check-all-that-apply (CATA) test, the evaluators were questioned about the identification of 24 sensory attributes. The terms were generated by a group of five researchers, who investigated the literature and tasted the yogurts [33]. The questionnaire terms were randomized and balanced among the evaluators [34]. For the ranking test, panelists ranked the samples in terms of overall preference (from most favorite to least favorite), following Meilgaard et al. [35]. The project was evaluated and approved by the Research Ethics Committee of the Alcides Carneiro University Hospital of the Federal University of Campina Grande under CAAE: 50775921.7.0000.5182. It was approved on 16 September 2021.

### 2.11. Statistical Analysis

All determinations were performed in triplicate (*n* = 3). The data were analyzed using Statistica software, version 13.0, Tulsa, Oklahoma, USA (StatSoft), and assessed through analysis of variance (ANOVA). Means were compared using Tukey’s test, considering a significance level of 5% (*p* < 0.05). The results of the CATA test were processed using XLSTAT-Sensory^®^ Software 2023.3 (Addinsoft). Significant differences between the samples for each term were evaluated using Cochran’s Q test. The chi-squared test was applied to verify the association of attributes among different samples. Additionally, correspondence analysis (CA) was conducted to obtain a two-dimensional representation of the relationship between samples and mentioned attributes; CA distances were calculated using chi-squared tests. For all statistical tests, a significance level of 5% was used. For the ranking test, the existence of differences between the samples was evaluated using Friedman’s test; subsequently, the differences between the samples were determined according to Christensen et al. [36], both considering a significance level of 5%.

## 3. Results and Discussion

### 3.1. Physical and Physicochemical Properties of the Prepared Yogurts

The results obtained in Table 1 present the pH and acidity values of the yogurts prepared with different concentrations of *Malvaviscus arboreus* flower and leaf extract (FE and LE) analyzed over a 28-day storage period. A significant reduction (*p* < 0.05) in acidity levels was observed in the YLE2%, YFE1%, and YFE2% treatments over the 28-day storage period. These findings might be attributed to the production of lactic acid by the lactic acid bacteria present in the yogurt during the storage period. Similar observations were reported for yogurt supplemented with *Phoenix dactylifera* L. extract [37] and yogurt enriched with rose flower extract (*Rosa rugosa* cv. Plena) [38]. Additionally, lower acidity in yogurts is beneficial as it is associated with better consumer acceptance [39].

Significant differences in Aw were observed among the formulations, particularly on day 14. For example, YLE2% and YFE2% exhibited lower Aw compared to the control (YC) (*p* < 0.05). The lower Aw in YLE2% and YFE2% can be attributed to the interaction of phenolic compounds with water molecules, which may improve the water-binding capacity and reduce the free water availability. This behavior enhances microbial stability and prolongs shelf life, which is an essential characteristic of functional dairy products.

The decrease in acidity and pH observed in the yogurt formulations containing *Malvaviscus arboreus* flower and leaf extracts (FE and LE) during the 28-day storage period is an interesting phenomenon. However, a concomitant decrease in both parameters is not typical. This suggests that other factors, such as the interaction between the malvaviscus extracts and the yogurt matrix, may influence acidity and pH levels. The extracts may contain compounds that buffer or neutralize some of the acids produced by the LAB, leading to a reduction in measured acidity without a significant drop in pH.

The yogurts’ lactose content and total carbohydrates were significantly reduced (*p* < 0.05) in all samples over the storage period. This reduction might be associated with the probiotic culture’s metabolism since lactose is the primary carbohydrate source for lactic acid bacteria [40]. However, the YC sample obtained lower concentrations of lactose (3.66 g/100 g) and total carbohydrates (10.15 g/100 g) at the end of the 28-day storage, suggesting a reduction in fermentative activity in treatments supplemented with FE and LE. This effect could be justified by the inhibition of starter cultures by the extract, as observed in the viability results of lactic acid bacteria. Asensio-Vegas et al. [41] reported that the starter culture used in this study has a short latency phase, leading to rapid fermentation with high lactose hydrolysis. However, Fang and Guo [20] and Xanthopoulos et al. [42] have described higher lactose values in goat yogurts than those in this study.

The inhibition of starter cultures by *Malvaviscus* extracts may not be entirely negative, as it could contribute to the desired characteristics of the final product, such as a milder flavor. However, further investigation would be needed to fully understand the impact of this inhibition on the probiotic properties and overall functional benefits of the yogurt.

The results presented in Table 1 demonstrate that the moisture content of all yogurt formulations remained above 79 g/100 g throughout the 28-day storage period. However, it was evident that the inclusion of extracts in yogurt significantly reduced moisture content during storage (*p* < 0.05). It is noteworthy that moisture content plays a crucial role in the quality of dairy products, influencing various characteristics such as physical, physicochemical, microbiological, and textural properties. These moisture alterations can consequently affect technological transformations and the storage of the final product [43].

The results revealed a consistent increase in the lipid content at the end of the storage period (28 days) for all evaluated formulations (Table 1). The recorded values were 5.47 g/100 g (YLE1%), 5.44 g/100 g (YLE2%), 5.48 g/100 g (YFE1%), and 5.54 g/100 g (YFE2%). We associate this behavior with a possible redistribution of yogurt matrix components, including lipids, during storage. There is likely a migration of lipids from other structures present in yogurt, resulting in a significant increase [44]. Similar results regarding lipid content changes in yogurts during storage have been described by El-Shibiny et al. [45] and Bezerril et al. [46]. Between the formulations, YLE2% demonstrated a greater moisture content reduction, contributing to its higher Aw and slightly lower lipid redistribution. Conversely, YFE2% maintained better moisture retention and achieved the highest lipid content at day 28, enhancing its firmness and reducing syneresis. This behavior highlights the distinct functional roles of the extracts, with the flower extract (FE) favoring better matrix stability and texture.

The highest protein content at the end of storage was observed in YLE2% and YFE1% (*p* < 0.05). This suggests that these formulations benefit from interactions between the extracts and milk proteins, stabilizing the protein network.

The extracts increased the ash content on the first day (*p* < 0.05). However, by the end of storage, no significant differences were found between the samples (*p* > 0.05). Contradictory results were reported by El-Sayed et al. [47] when evaluating yogurts fortified with different concentrations of zinc nanoparticles, in which they observed an increase in ash content in supplemented samples at the end of storage. The YLE2% and YFE1% formulations achieved the highest protein contents at the end of the 28-day storage period (Table 1) (*p* < 0.05). The YLE2% and YFE1% formulations achieved the highest protein contents at the end of the 28-day storage period (Table 1) (*p* < 0.05). Similar results were reported for goat yogurts supplemented with xique-xique flour [4]. Food products need to be appealing to be accepted by consumers [15], as the product’s appearance influences them; thus, color is one of the most relevant appearance aspects [48].

The formulations with flower extract (YFE1% and YFE2%) demonstrated higher energy values than those with leaf extract (YLE1% and YLE2%) and the control. This difference is primarily due to the higher lipid content in YFE1% and YFE2%, which increased significantly during storage (*p* < 0.05). YFE2%, with the highest energy value, reflects the enhanced lipid redistribution promoted by the flower extract. In contrast, YLE2% showed a lower energy value, likely due to its comparatively lower lipid content.

These results emphasize the diverse impact of malvaviscus extracts on yogurt formulations. The flower extract, especially at 2%, enhanced lipid redistribution and energy value while improving functional properties such as firmness and reducing syneresis. In contrast, the leaf extract influenced water activity and protein content, providing unique textural characteristics. This study illustrates how varying extract types and concentrations can strategically tailor yogurt properties for specific consumer preferences and functional benefits by comparing the five formulations.

Table 2 displays the results of the yogurt color analysis during storage. Adding FE and LE from the malvaviscus influenced most of the instrumental color parameters of the yogurts (*p* < 0.05). A higher concentration (2%) of malvaviscus FE increased the expression of redness (+a*) and reduced its luminosity (L*) at Time 1, manifesting a pink color. Over the storage period, there was a loss of coloration. This behavior might be associated with the oxidation of phenolic compounds, which are sensitive to oxidation [49]. The oxidation of polyphenols is a well-known cause of color degradation in foods, including dairy products, as it alters the phenolic compounds that contribute to the color.

Additionally, the pH of the yogurt plays a crucial role in the stability of the polyphenolic compounds. A decrease in pH, which is common in yogurt due to fermentation, can further accelerate the oxidation process or influence polyphenols’ solubility and color stability. As the pH of the yogurt decreases, the phenolic compounds may become more susceptible to oxidative changes, leading to a gradual loss of color intensity. However, it was the treatment with the highest a* value compared to others (*p* < 0.05). These results align with the findings of Sohrabpour et al. [50], who discovered higher a* values for yogurt supplemented with aqueous cinnamon extract and reduced L* values with increased extract concentration.

### 3.2. Bioactive Compounds and Total Antioxidant Activity of Developed Goat Yogurts

At the end of the storage period, the addition of malvaviscus LE and FE to goat yogurts significantly increased (*p* < 0.05) the levels of phenolic compounds and antioxidant activity (ABTS and FRAP assays), as shown in Table 3. The promoted antioxidant activity is associated with the high content of phenolic compounds present in the extracts [51,52]. These results align with the existing literature, in which adding plant extracts has been observed to increase antioxidant activity in yogurts [53,54,55]. It is important to note that probiotic yogurts have antioxidant activity due to bioactive peptides [56]. Hence, it is possible to quantify these substances even in samples without extract addition. This effect has been previously reported in the literature [15,57].

There was a significant reduction in phenolic compounds, flavonoids, and antioxidant activity levels throughout the storage period in all treatments. However, the formulations supplemented with extracts promoted a higher concentration of these substances, except for the content of phenolic compounds at the end of day 28. The degradation of antioxidant compounds in yogurts occurs during storage, as observed previously [58,59]. One reason for this occurrence is the potential binding between polyphenols and yogurt proteins [60]. Another explanation is their involvement in combating lipid peroxidation, as phenolic antioxidants can donate hydrogen atoms to radicals, rendering them relatively stable [61]. The YFE2% sample exhibited superior antioxidant activity compared to the others during storage. This result was expected because the sample had a higher extract concentration with greater antioxidant activity (Appendix A).

### 3.3. Lipid Oxidation in Produced Goat Yogurts

Lipid oxidation is a significant challenge for researchers and the food industry [62]. During the oxidative process, unwanted compounds form, resulting in unpleasant sensory characteristics and adverse effects on human health [63]. Furthermore, the milk fat globule membrane is primarily formed by phospholipids and can be a primary focus of oxidation [64].

The FE and LE malvaviscus significantly delayed (*p* < 0.05) lipid oxidation in yogurts during storage (Table 4). Treatments containing FE (YFE1% and YFE2%) showed higher oxidative stability compared to others (*p* < 0.05). As expected, a higher oxidation index was observed in all treatments at the end of the storage period (28 days). The YC treatment significantly differed (*p* < 0.05) from all other prepared yogurts, showing higher oxidation from the 14th day of storage. This result might be explained by the antioxidant activity of the extracts, which promoted more excellent stability at the end of storage. This phenomenon was previously observed when analyzing peroxidation in yogurts supplemented with plant extract [65].

### 3.4. Sanitary Quality and Viability of Lactic Acid Bacteria in Produced Goat Yogurts

The results obtained from the microbiological analysis of the hygienic–sanitary quality of yogurts indicated that all developed formulations were suitable for human consumption throughout the refrigerated storage period, meeting the current Brazilian legislation standards [27]. Table 5 shows the viability of lactic acid bacteria during refrigerated storage. It was observed that adding extracts affected the viability of *L. bulgaricus* and *L*. *casei* (*p* < 0.05), suggesting that the extracts acted as preservatives by inhibiting microbiological growth, including lactic acid strains. Previous studies have described the antimicrobial activity [12] and antibacterial properties [66] of malvaviscus leaves and flowers. Throughout storage, there was a reduction in the counts of starter and probiotic bacteria (*p* < 0.05). By the end of storage, the counts of *L. delbrueckii* subsp. *bulgaricus* in YFE1%, YFE2%, and YLE1% averaged < 1.0, while the YLE2% and YC treatments did not show significant differences (*p* > 0.05) between them.

The counts of S. thermophilus during storage ranged from 7.84 ± 0.06 log CFU/g at the beginning to 4.93 ± 0.02 log CFU/g at the end of the storage period. By day 28, all yogurt treatments containing malvaviscus extract exhibited significantly lower counts (*p* < 0.05) than the control. This reduction may be attributed to the antimicrobial properties of the malvaviscus extracts, which could interact with microbial viability during extended storage. However, for the probiotic culture L. casei, extract-added samples showed higher viable cell counts at the end of storage (*p* < 0.05), albeit with values not exceeding 5.24 log CFU/g. The decline in microbial viability, especially in starter cultures, is likely linked to the production of bioactive substances by lactic acid bacteria, such as organic acids, hydrogen peroxide, antimicrobial peptides, and other microbial metabolites [18,67]. Additionally, the low pH in fermented products enhances the levels of undissociated organic acids, which are known to expand bactericidal activity [68]. As observed in previous studies, storage and post-acidification further contribute to this decline [69,70,71,72].

Interestingly, *L. casei* demonstrated better resilience than starter cultures, likely due to its resistance to low pH and post-acidification [73]. While the recommended viable cell count for probiotic benefits is estimated to range between 6.0 and 7.0 log CFU/100 g [74], recent research suggests that viability is not a strict determinant of health benefits. Non-viable, inactivated, or even ruptured probiotic cells and their metabolites have shown therapeutic potential in addressing various pathologies [75,76,77]. These findings highlight the need to explore the therapeutic properties of yogurt further, extending the focus beyond viable cell counts to the broader spectrum of bioactive contributions.

### 3.5. Texture Profile and Syneresis of Elaborated Goat Yogurts

The textural properties of yogurt gels and syneresis are critical indicators of yogurt quality and are closely linked to sensory acceptance [78]. Over the 28-day storage period, texture profile and syneresis analyses (Table 6) demonstrated that the addition of *Malvaviscus* flower extract (FE) significantly enhanced yogurt firmness (*p* < 0.05), particularly at the end of storage. This increase in firmness can be attributed to polyphenol–protein interactions, which likely lead to the formation of complexes with milk proteins, such as casein. These interactions stabilize the yogurt matrix, improving texture and viscosity [53,79].

These findings are consistent with previous studies, which also reported increased firmness in yogurts supplemented with plant extracts [80,81]. The ability of plant-derived polyphenols to interact with dairy proteins highlights their role not only as bioactive compounds but also as functional ingredients that enhance the structural integrity of the yogurt. This improved firmness contributes to the overall sensory quality, making yogurts with FE additions more desirable for consumers.

The addition of malvaviscus leaf (LE) and flower extracts (FE) significantly influenced the resilience of yogurt, particularly at the beginning of storage. This effect is closely correlated with the higher firmness values observed in the yogurts supplemented with the extracts at this stage (Table 6). Furthermore, the resilience increased significantly (*p* < 0.05) in the yogurt formulations with malvaviscus extracts between Day 1 and Day 28, aligning with studies that report enhanced resilience in protein-enriched yogurts [82] and those supplemented with modified starches, such as succinylated octenyl millet starch (Pennisetum typhoides) [83,84].

FE also led to a marked increase (*p* < 0.05) in cohesiveness from Day 1 to Day 28. This behavior suggests that the phenolic compounds in the extract interact with yogurt proteins, particularly casein, promoting rearrangement during storage. These interactions stabilize the protein network, enhancing yogurt consistency and forming softer gels [85,86,87]. Similarly, adding LE and FE improved elasticity on Day 1, with YLE2%, YFE1%, and YFE2% yogurts demonstrating significantly higher elasticity (*p* < 0.05). These results are consistent with prior findings, such as the work by Żbikowska et al. [88], in which inulin supplementation increased yogurt elasticity. Similarly, Mousavi et al. [89] reported improvements in texture and sensory characteristics in yogurt supplemented with flaxseed during cold storage, highlighting the potential of dietary fiber additives to enhance yogurt quality.

Gumminess, which is an undesirable characteristic of yogurt texture, was reduced (*p* < 0.05) in samples supplemented with leaf extracts and 1% flower extract by the end of storage. This contributed to softer gels, which is a trend that is also observed in yogurts fortified with Algerian leaf extract (*Solenostemma algel* Hayne) [90]. While the extracts initially promoted greater chewiness (*p* < 0.05), YFE2% and YC maintained higher chewiness at the storage end, which is consistent with YFE2%’s superior firmness.

Syneresis, which is a key indicator of yogurt quality, reflects the matrix’s water-binding capacity and impacts consumer acceptability [91]. The syneresis values in this study ranged from 16.6 ± 0.82% to 21.31 ± 1.63%, which is considerably lower than values reported for yogurts with plant seed mucilage (70–80%) [92,93] or other plant extracts (35–50%) [94,95]. This is favorable, as high syneresis negatively affects yogurt acceptability. The YFE2% sample exhibited the lowest whey release rate (*p* < 0.05) by the end of storage, which is a result attributed to its greater firmness, as firmer yogurts are less prone to syneresis [79,96,97]. This underscores the potential of malvaviscus extracts, particularly FE, in improving yogurt texture and reducing whey separation, enhancing both product quality and consumer satisfaction.

### 3.6. Rheological Behavior of Elaborated Goat Yogurts

The results of the Herschel–Bulkley model fitting showed excellent accuracy, with R^2^ values ≥ 99.36% during the storage period (Appendix A). This model is well-established for describing the rheological behavior of yogurt [67,98]. Regression coefficients obtained for the model at various storage temperatures are provided in Appendix A. The consistency index (*k*), which is a key indicator of yogurt viscosity [99], increased significantly from the 14th day of storage in samples containing malvaviscus extracts at all tested temperatures. By the end of storage, yogurts with flower extract (FE) exhibited higher *k* values than other formulations, signifying enhanced thickening properties. These findings are consistent with the textural data presented in Table 5, which showed increased firmness and cohesiveness in FE-supplemented samples.

Such changes in rheological parameters during storage are expected in yogurt products [100]. Similar trends have been reported by Barukčić et al. [101], who observed a rising consistency index over time in yogurts enriched with olive leaf extract. This phenomenon can be attributed to the hydration of macromolecules, which strengthens the yogurt’s three-dimensional network, and the stabilizing effects of the plant-based ingredients [93,102]. Polyphenol–protein interactions likely contribute to these changes by forming complexes that enhance the yogurt’s viscosity and improve its overall texture profile [87]. These findings highlight malvaviscus extracts’ functional benefits in enhancing yogurt’s rheological stability and quality during extended storage.

Furthermore, an increase in the flow behavior index was observed with the addition of extracts. This phenomenon is due to the aggregation of the casein network through interaction with phenolic compounds, which reduces the resistance of yogurt to flow [103,104]—in a general assessment of fluid behavior, temperatures and storage time influenced the rheological behavior of yogurts. Additionally, it was observed that shear stress is dependent on shear rate because, with an increase in shear stress, there is an increase in yogurt deformation rate. These results are consistent with reports previously described for yogurts [99,105].

### 3.7. Sensory Analysis

The sensory attributes evaluated in the yogurt samples are summarized in Appendix A. Among the 24 sensory terms assessed, only one attribute—acidic aroma—did not show a statistically significant difference among the formulations (*p* > 0.05). This finding highlights the importance of the remaining characteristics in defining and differentiating the sensory profiles of the yogurts. Correspondence analysis (CA) was used to illustrate the relationship between samples and attributes, with the first two dimensions explaining 97.90% of the total inertia (Figure 1).

Samples YFE1% and YFE2% were associated with attributes such as pink color, bright appearance, floral aroma, characteristic yogurt flavor, and soft texture, which resonated positively with consumer preferences. These sensory traits contributed to the higher acceptance of YFE2%, as reflected in the preference ranking (Table 7). In contrast, samples YLE1% and YLE2% were linked to less favorable attributes, including bitterness, herbal taste, green color, aftertaste, viscosity, and opacity. These characteristics likely contributed to their lower consumer preference. Similar findings have been reported in studies in which plant leaf extracts reduced the sweetness and overall acceptance of yogurts [106,107].

These results demonstrate the significant influence of malvaviscus flower and leaf extracts on the sensory properties of goat yogurt. While the flower extract enhanced sensory appeal through attributes like color and flavor, the leaf extract introduced less desirable characteristics, indicating that the type and concentration of plant-based ingredients play a critical role in determining consumer preference.

The ranking test showed that yogurt YFE2% was preferred by the evaluators (*p* < 0.05), while YC and YFE1% did not show significant differences between them (Table 7). Coloration may have influenced this aspect since yogurts with a more intense pink color have higher acceptability [108]. Overall, adding 2% of FE improved the sensory characteristics of goat yogurts. These results are similar to those found in yogurt supplemented with flower extract [109].

## 4. Conclusions

This study demonstrated that *Malvaviscus* leaf (LE) and flower extracts (FE) hold considerable potential as functional ingredients in goat yogurt production. Incorporating these extracts significantly enhanced the antioxidant activity of the yogurts and effectively inhibited lipid oxidation during extended storage (*p* < 0.05). Even at low concentrations, the extracts influenced yogurt coloration, making it distinguishable by the sensory panel and contributing to its visual appeal. In addition to their antioxidant benefits, the extracts improved the yogurts’ key sensory, textural, and syneresis properties. Notably, the formulation with 2% flower extract (YFE2%) exhibited the best overall performance, with the highest sensory preference, the lowest syneresis percentage at the end of storage, and a firmer texture. These results highlight the potential for *Malvaviscus* extracts, particularly FE, to enhance goat yogurts’ functional and sensory quality. This research provides valuable insights into developing antioxidant-rich dairy products with improved consumer appeal.

## Figures and Tables

**Figure 1 foods-13-03942-f001:**
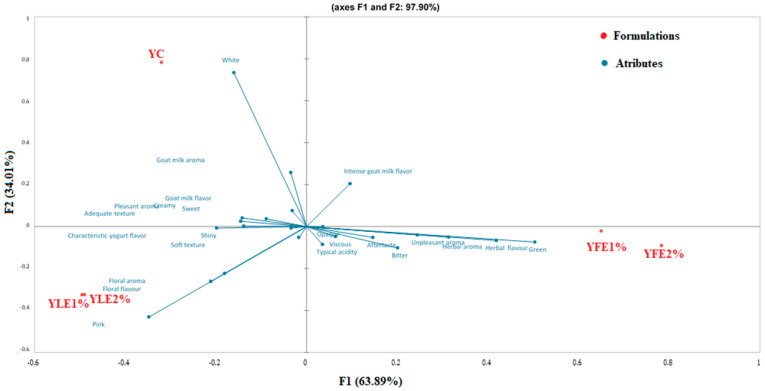
Correspondence analysis of terms elicited from the CATA test.

**Table 1 foods-13-03942-t001:** Physical and physico-chemical characterization of goat yogurts prepared with the addition of malvaviscus flower extract (FE) and malvaviscus leaf extract (LE) during 28 days of refrigerated storage.

Parameters	Days	Formulations				
		YC	YLE1%	YLE2%	YFE1%	YFE2%
Aw	1	0.996 ± 0.00 ^bA^	0.997 ± 0.00 ^aA^	0.994 ± 0.00 ^cA^	0.994 ± 0.00 ^cA^	0.994 ± 0.00 ^cA^
	14	0.995 ± 0.00 ^aA^	0.996 ± 0.00 ^aA^	0.988 ± 0.00 ^bB^	0.989 ± 0.00 ^bB^	0.983 ± 0.00 ^cB^
	28	0.992 ± 0.00 ^cB^	0.995 ± 0.00 ^bA^	0.995 ± 0.00 ^bA^	0.997 ± 0.00 ^aA^	0.996 ± 0.00 ^aA^
Lactic acid acidity (g/100 g)	1	0.88 ± 0.00 ^dA^	0.95 ± 0.01 ^bA^	0.91 ± 0.01 ^cA^	0.95 ± 0.01 ^bA^	1.02 ± 0.00 ^aA^
	14	0.84 ± 0.00 ^cB^	0.83 ± 0.01 ^cB^	0.84 ± 0.02 ^cB^	0.91 ± 0.00 ^bB^	0.97 ± 0.01 ^aB^
	28	0.89 ± 0.01 ^cA^	0.95 ± 0.01 ^bA^	0.85 ± 0.01 ^dB^	0.93 ± 0.01 ^bAB^	0.98 ± 0.00 ^aB^
pH	1	4.7 ± 0.00 ^aA^	4.6 ± 0.00 ^bA^	4.6 ± 0.00 ^bA^	4.7 ± 0.00 ^aA^	4.7 ± 0.06 ^aA^
	14	4.4 ± 0.16 ^aB^	4.3 ± 0.06 ^abB^	4.2 ± 0.06 ^bB^	4.2 ± 0.16 ^bB^	4.2 ± 0.16 ^bB^
	28	4.4 ± 0.06 ^aB^	4.3 ± 0.16 ^abB^	4.2 ± 0.06 ^bB^	4.1 ± 0.06 ^bB^	4.1 ± 0.00 ^bB^
Lactose (g/100 g)	1	3.81 ± 0.04 ^aA^	3.86 ± 0.02 ^aA^	3.86 ± 0.02 ^aA^	3.89 ± 0.04 ^aA^	3.88 ± 0.06 ^aA^
	14	3.64 ± 0.02 ^bB^	3.69 ± 0.02 ^aC^	3.74 ± 0.04 ^aB^	3.74 ± 0.04 ^aB^	3.75 ± 0.04 ^aB^
	28	3.66 ± 0.04 ^bB^	3.76 ± 0.02 ^aB^	3.79 ± 0.02 ^aB^	3.74 ± 0.00 ^aB^	3.75 ± 0.02 ^aB^
Carbohydrates (g/100 g)	1	10.94 ± 0.22 ^bA^	12.17 ± 0.39 ^bA^	12.64 ± 0.29 ^aA^	13.04 ± 0.18 ^aA^	13.00 ± 0.28 ^aA^
	14	11.20 ± 0.11 ^bA^	12.36 ± 0.28 ^aA^	12.77 ± 0.22 ^aA^	12.82 ± 0.26 ^aA^	13.10 ± 0.57 ^aA^
	28	10.15 ± 0.38 ^cB^	11.05 ± 0.87 ^bA^	11,76 ± 0.90 ^aB^	11.50 ± 0.08 ^abB^	10.80 ± 0.56 ^bB^
Moisture (g/100 g)	1	82.59 ± 0.17 ^aA^	81.25 ± 0.04 ^bA^	81.27 ± 0.03 ^bA^	81.09 ± 0.16 ^bA^	81.21 ± 0.13 ^bA^
	14	82.64 ± 0.07 ^aA^	81.32 ± 0.26 ^bA^	81.27 ± 0.11 ^bA^	81.32 ± 0.19 ^bA^	81.44 ± 0.43 ^bA^
	28	81.19 ± 0.24 ^aB^	79.67 ± 0.26 ^cB^	78.67 ± 0.97 ^bcB^	79.15 ± 0.17 ^bcB^	79.93 ± 0.69 ^acB^
Lipids (g/100 g)	1	2.64 ± 0.03 ^aB^	2.47 ± 0.02 ^bB^	2.11 ± 0.02 ^cB^	2.05 ± 0.03 ^dB^	1.94 ± 0.03 ^eB^
	14	2.51 ± 0.03 ^aB^	2.52 ± 0.03 ^aB^	2.12 ± 0.03 ^bB^	2.05 ± 0.03 ^bB^	1.94 ± 0.03 ^cB^
	28	4.85 ± 0.23 ^bA^	5.47 ± 0.25 ^aA^	5.44 ± 0.08 ^aA^	5.48 ± 0.06 ^aA^	5.54 ± 0.27 ^aA^
Ashes (g/100 g)	1	0.69 ± 0.01 ^bB^	0.76 ± 0.02 ^aA^	0.75 ± 0.04 ^aA^	0.73 ± 0.01 ^abB^	0.72 ± 0.01 ^abA^
	14	0.69 ± 0.01 ^cB^	0.74 ± 0.02 ^abA^	0.77 ± 0.01 ^aA^	0.71 ± 0.01 ^bB^	0.73 ± 0.02 ^bA^
	28	0.74 ± 0.01 ^aA^	0.76 ± 0.01 ^aA^	0.74 ± 0.02 ^aA^	0.75 ± 0.01 ^aA^	0.73 ± 0.02 ^aA^
Protein (g/100 g)	1	3.13 ± 0.13 ^abA^	3.34 ± 0.34 ^aA^	3.12 ± 0.25 ^abAB^	3.10 ± 0.04 ^bAB^	3.12 ± 0.19 ^bA^
	14	2.96 ± 0.08 ^abB^	3.06 ± 0.12 ^abB^	3.07 ± 0.14 ^abB^	3.10 ± 0.10 ^aB^	2.79 ± 0.11 ^bC^
	28	3.06 ± 0.06 ^bA^	3.06 ± 0.06 ^bB^	3.38 ± 0.39 ^aA^	3.12 ± 0.06 ^abA^	2.99 ± 0.05 ^bB^
Energetic value (kcal/100 g)	1	80.08 ± 0.79 ^cB^	84.28 ± 0.07 ^aB^	82.39 ± 0.20 ^bB^	82.94 ± 0.54 ^abB^	81.98 ± 0.67 ^bB^
	14	79.18 ± 0.43 ^cB^	84.32 ± 1.05 ^aB^	82.44 ± 0.42 ^abB^	82.12 ± 0.86 ^abB^	81.01 ± 1.62 ^bcB^
	28	96.52 ± 0.81 ^bA^	105.65 ± 1.37 ^aA^	109.55 ± 3.57 ^aA^	107.76 ± 0.98 ^aA^	105.07 ± 3.97 ^aA^

Note: Aw: water activity; YC: control goat yogurt; YLE1%: goat yogurt with 1% addition of LE; YLE2%: goat yogurt with 2% addition of LE; YFE1%: goat yogurt with 1% addition of FE; YFE2%: goat yogurt with 2% addition of F; ^a–e^ mean ± standard deviation with different lowercase letters on the same line differed by Tukey’s test (*p* < 0.05) between treatments; ^A–C^ mean ± standard deviation with different capital letters in the same column differed by Tukey’s test (*p* < 0.05), over storage time.

**Table 2 foods-13-03942-t002:** Chromatic coordinates (L*, a*, and b*) of goat yogurts prepared with different concentrations (1 and 2%) of malvaviscus flower extract (FE) and malvaviscus leaf extract (LE) during 28 days of refrigerated storage.

Parameters	Days	Formulations				
		YC	YLE1%	YLE2%	YFE1%	YFE2%
L*	1	45.36 ± 1.68 ^aA^	45.34 ± 0.99 ^aA^	43.79 ± 1.89 ^abA^	42.45 ± 0.71 ^bA^	38.67 ± 0.90 ^cA^
	14	32.82 ± 0.89 ^cB^	39.80 ± 0.71 ^aB^	36.68 ± 0.91 ^bB^	30.44 ± 0.45 ^dB^	25.32 ± 0.69 ^eC^
	28	32.29 ± 0.92 ^aB^	32.33 ± 0.51 ^aC^	32.77 ± 0.77 ^aC^	28.15 ± 0.56 ^bC^	28.68 ± 0.83 ^bB^
a*	1	−2.07 ± 0.72 ^cA^	−1.79 ± 0.51 ^cA^	−2.45 ± 0.36 ^cA^	−0.13 ± 0.86 ^bA^	1.48 ± 0.98 ^aA^
	14	−3.65 ± 0.82 ^cB^	−3.07 ± 0.80 ^cB^	−3.55 ± 0.71 ^cB^	−1.08 ± 0.54 ^bB^	0.92 ± 0.82 ^aA^
	28	−3.13 ± 0.79 ^dB^	−2.58 ± 0.74 ^bdB^	−3.77 ± 0.89 ^cB^	−1.81 ± 0.64 ^bB^	−0.12 ± 0.64 ^aB^
b*	1	12.67 ± 0.96 ^bC^	12.60 ± 0.98 ^bA^	14.33 ± 0.99 ^aC^	11.82 ± 0.94 ^bC^	11.50 ± 0.97 ^bB^
	14	17.81 ± 0.56 ^aA^	13.73 ± 0.59 ^cA^	17.55 ± 0.99 ^aA^	15.07 ± 0.79 ^bB^	17.27 ± 0.91 ^aA^
	28	14.21 ± 0.42 ^cB^	13.74 ± 1.38 ^cA^	16.09 ± 0.97 ^bB^	22.96 ± 1.40 ^aA^	11.69 ± 0.52 ^dB^

Note: YC: control goat yogurt; YLE1%: goat yogurt with 1% addition of LE; YLE2%: goat yogurt with 2% addition of LE; YFE1%: goat yogurt with 1% addition of FE; YFE2%: goat yogurt with 2% addition of FE; ^a–e^ mean ± standard deviation with different lowercase letters on the same line differed by Tukey’s test (*p* < 0.05), between treatments; ^A–C^ mean ± standard deviation with different capital letters in the same column differed by Tukey’s test (*p* < 0.05), over storage time.

**Table 3 foods-13-03942-t003:** Phenolic compounds, flavonoids, and antioxidant activity (FRAP and ABTS) of goat yogurts prepared with malvaviscus flower extract (YLE1% and YLE2%) and malvaviscus leaf extract (YFE1% and YFE2%) during 28 days of refrigerated storage.

Parameters	Days	Formulations				
		YC	YLE1%	YLE2%	YFE1%	YFE2%
Phenolic compounds (mg GAE/100 g)	1	7.98 ± 0.19 ^cA^	11.57 ± 0.19 ^aA^	6.16 ± 0.11 ^dA^	5.97 ± 0.11 ^dA^	9.68 ± 0.19 ^bA^
	14	7.94 ± 0.07 ^cA^	11.55 ± 0.18 ^aA^	6.15 ± 0.10 ^dA^	6.00 ± 0.08 ^dA^	9.66 ± 0.17 ^bA^
	28	5.53 ± 0.19 ^bB^	8.30 ± 0.48 ^aB^	3.39 ± 0.39 ^cB^	3.83 ± 0.19 ^cB^	8.23 ± 0.48 ^aB^
Flavonoids (mg CE/100 g)	1	<LOD	0.39 ± 0.00 ^dA^	0.41 ± 0.01 ^cA^	0.45 ± 0.00 ^bA^	0.51 ± 0.01 ^aA^
	14	<LOD	<LOD	0.40 ± 0.00 ^bB^	0.39 ± 0.00 ^bB^	0.45 ± 0.01 ^aB^
	28	<LOD	<LOD	0.40 ± 0.01 ^aB^	<LOD	0.39 ± 0.00 ^aC^
FRAP (µmol TEAC/g)	1	0.41 ± 0.01 ^eA^	0.54 ± 0.01 ^cB^	0.49 ± 0.01 ^dA^	0.71 ± 0.00 ^bA^	0.89 ± 0.01 ^aA^
	14	0.40 ± 0.00 ^cA^	0.67 ± 0.23 ^bA^	0.49 ± 0.01 ^cA^	0.70 ± 0.00 ^bB^	0.87 ± 0.00 ^aA^
	28	0.38 ± 0.00 ^eB^	0.52 ± 0.00 ^cB^	0.45 ± 0.00 ^dB^	0.66 ± 0.00 ^bC^	0.78 ± 0.00 ^aB^
ABTS (µmol TEAC/g)	1	3.97 ± 0.16 ^cA^	4.16 ± 0.06 ^cA^	4.23 ± 0.07 ^cA^	8.88 ± 0.08 ^bA^	10.17 ± 0.29 ^aA^
	14	2.09 ± 0.08 ^dB^	3.84 ± 0.03 ^cB^	3.89 ± 0.06 ^cB^	4.80 ± 0.21 ^bB^	9.19 ± 0.07 ^aB^
	28	1.94 ± 0.05 ^dB^	2.98 ± 0.02 ^cC^	3.11 ± 0.12 ^cC^	3.42 ± 0.16 ^bC^	8.52 ± 0.12 ^aC^

Note: YC: control goat yogurt; YLE1%: goat yogurt with 1% addition of LE; YLE2%: goat yogurt with 2% addition of LE; YFE1%: goat yogurt with 1% addition of FE; YFE2%: goat yogurt with 2% addition of FE; ^a–e^ mean ± standard deviation with different lowercase letters on the same line differed by Tukey’s test (*p* < 0.05), between treatments. ^A–C^ mean ± standard deviation with different capital letters in the same column differed by Tukey’s test (*p* < 0.05) over storage time; <LOD: below the detection limit.

**Table 4 foods-13-03942-t004:** Mean TBARS values of goat yogurts prepared with different concentrations (1 and 2%) of malvaviscus flower (FE) and leaf (LE) extracts during 28 days of refrigerated storage.

Days	Formulations				
	YC	YLE1%	YLE2%	YFE1%	YFE2%
1	0.017 ± 0.00 ^bB^	0.017 ± 0.00 ^bC^	0.021 ± 0.00 ^aC^	0.021 ± 0.00 ^aB^	0.017 ± 0.00 ^bB^
14	0.130 ± 0.01 ^aB^	0.109 ± 0.00 ^bB^	0.109 ± 0.00 ^bB^	0.026 ± 0.01 ^cB^	0.021 ± 0.01 ^cB^
28	0.161 ± 0.00 ^aA^	0.125 ± 0.00 ^cA^	0.146 ± 0.00 ^bA^	0.047 ± 0.00 ^eA^	0.078 ± 0.00 ^dA^

Note: Results are expressed as average (n = 3) ± standard deviation. The values correspond to mg MDA/kg of sample; ^a–e^ lowercase letters differing within the same row were significantly different according to Tukey’s test (*p* < 0.05) (between the treatments); ^A–C^ capital letters differing within the same column were significantly different according to Tukey’s test (*p* < 0.05) (between the storage time); YC: control goat yogurt; YLE1%: goat yogurt with 1% addition of LE; YLE2%: goat yogurt with 2% addition of LE; YFE1%: goat yogurt with 1% addition of FE; YFE2%: goat yogurt with 2% addition of FE.

**Table 5 foods-13-03942-t005:** Viability of lactic acid bacteria in goat yogurts prepared with malvaviscus flower extract (YLE1% and YLE2%) and malvaviscus leaf extract (YFE1% and YFE2%) during 28 days of refrigerated storage.

Lactic Bacteria	Days	Formulations				
		YC	YLE1%	YLE2%	YFE1%	YFE2%
*L. bulgaricus*	1	7.90 ± 0.00 ^aA^	7.13 ± 0.02 ^bA^	6.44 ± 0.02 ^cA^	5.22 ± 0.01 ^dA^	5.16 ± 0.02 ^dA^
	14	5.53 ± 0.01 ^aB^	4.42 ± 0.05 ^bB^	5.37 ± 0.01 ^aB^	1.0 ± 0.01 ^cB^	<1.0 ± 0.01 ^cB^
	28	4.80 ± 0.02 ^aC^	<1.0 ± 0.58 ^bC^	4.96 ± 0.01 ^aC^	<1.0 ± 0.58 ^bB^	<1.0 ± 0.58 ^bB^
*S. thermophilus*	1	7.80 ± 0.02 ^dA^	7.18 ± 0.02 ^cA^	7.04 ± 0.03 ^dA^	7.31 ± 0.01 ^bA^	7.84 ± 0.06 ^aA^
	14	6.36 ± 0.00 ^dB^	6.58 ± 0.03 ^bB^	6.11 ± 0.01 ^eB^	6.47 ± 0.00 ^cB^	6.93 ± 0.02 ^aB^
	28	6.18 ± 0.01 ^aC^	5.50 ± 0.00 ^bC^	5.41 ± 0.00 ^cC^	5.14 ± 0.00 ^dC^	4.93 ± 0.02 ^eC^
*L. casei*	1	7.49 ± 0.00 ^aA^	7.20 ± 0.01 ^bA^	6.72 ± 0.04 ^cA^	5.10 ± 0.01 ^dA^	5.07 ± 0.01 ^dA^
	14	5.35 ± 0.01 ^bB^	5.38 ± 0.01 ^abB^	5.40 ± 0.01 ^aB^	4.44 ± 0.03 ^dB^	4.83 ± 0.01 ^cB^
	28	4.00 ± 0.00 ^dC^	5.24 ± 0.01 ^aC^	5.07 ± 0.01 ^bC^	4.17 ± 0.00 ^cC^	4.17 ± 0.00 ^cC^

Note: YC: control goat yogurt; YLE1%: goat yogurt with 1% addition of LE; YLE2%: goat yogurt with 2% addition of LE; YFE1%: goat yogurt with 1% addition of FE; YFE2%: goat yogurt with 2% addition of FE; ^a–e^ mean ± standard deiation with different lowercase letters on the same line differed by Tukey’s test (*p* < 0.05) between treatments; ^A–C^ mean ± standard deviation with different capital letters in the same column differed by Tukey’s test (*p* < 0.05) over storage time.

**Table 6 foods-13-03942-t006:** Instrumental texture profile and syneresis of yogurts during storage of goat yogurts prepared with malvaviscus flower extract (YLE1% and YLE2%) and malvaviscus leaf extract (YFE1% and YFE2%) during 28 days of refrigerated storage.

Parameters	Days	Formulations				
		YC	YLE1%	YLE2%	YFE1%	YFE2%
Firmness (g)	1	11.04 ± 0.83 ^bC^	18.86 ± 0.45 ^aA^	12.29 ± 0.71 ^bB^	19,36 ± 0.56 ^aA^	18.49 ± 0.55 ^aB^
	14	12.87 ± 0.37 ^bB^	19.44 ± 0.12 ^aA^	16.08 ± 0.15 ^bA^	19.55 ± 0.31 ^aA^	13.43 ± 0.38 ^bC^
	28	14.99 ± 0.75 ^bA^	14.01 ± 0.36 ^bB^	13.76 ± 0.83 ^bB^	18.81 ± 0.27 ^aB^	20.02 ± 0.19 ^aA^
Adhesiveness (g.s)	1	−44.52 ± 1.91 ^cB^	−32.51 ± 0.51 ^abB^	−29.73 ± 1.92 ^aB^	−31.93 ± 3.08 ^abA^	−35.47 ± 1.68 ^bB^
	14	−23.36 ± 1.96 ^aA^	−32.96 ± 0.31 ^dA^	−29.14 ± 0.78 ^bB^	−35.16 ± 1.08 ^cdB^	−31.70 ± 1.11 ^bdA^
	28	−25.97 ± 0.49 ^aA^	−30.41 ± 2.02 ^bA^	−27.92 ± 0.79 ^abA^	−33.27 ± 0.68 ^cA^	−29.77 ± 0.39 ^bA^
Resilience (%)	1	56.12 ± 0.95 ^dC^	60.39 ± 0.35 ^cB^	64.43 ± 0.63 ^aC^	61.78 ± 0.41 ^bcB^	62.64 ± 0.79 ^abB^
	14	72.90 ± 0.77 ^aA^	62.98 ± 0.91 ^cA^	70.85 ± 1.27 ^aA^	63.94 ± 0.19 ^cA^	66.28 ± 0.75 ^bA^
	28	67.98 ± 0.43 ^aB^	64.20 ± 0.56 ^bA^	67.84 ± 0.65 ^abB^	64.01 ± 0.26 ^cA^	66.63 ± 0.24 ^bA^
Cohesiveness	1	0.85 ± 0.01 ^bB^	0.87 ± 0.01 ^abB^	0.86 ± 0.01 ^aB^	0.87 ± 0.00 ^abA^	0.87 ± 0.01 ^abB^
	14	0.93 ± 0.04 ^aA^	0.88 ± 0.01 ^aA^	0.91 ± 0.07 ^aA^	0.88 ± 0.01 ^aA^	0.89 ± 0.00 ^aA^
	28	0.89 ± 0.00 ^aAB^	0.87 ± 0.01 ^bAB^	0.88 ± 0.00 ^abB^	0.85 ± 0.05 ^cB^	0.88 ± 0.00 ^abAB^
Elasticity (%)	1	89.03 ± 0.76 ^cB^	88.68 ± 0.58 ^cA^	91.62 ± 0.62 ^aA^	90.81 ± 0.66 ^abA^	89.51 ± 0.08 ^bcB^
	14	91.56 ± 0.40 ^abA^	88.02 ± 0.68 ^dB^	92.08 ± 0.43 ^aA^	89.83 ± 0.15 ^cAB^	90.49 ± 0.27 ^bcA^
	28	90.58 ± 0.52 ^aA^	88.27 ± 0.47 ^cAB^	89.97 ± 0.57 ^aB^	88.40 ± 0.75 ^bcB^	89.73 ± 0.42 ^acB^
Gumminess	1	10.73 ± 0.74 ^bC^	16.40 ± 0.30 ^aA^	11.99 ± 0.57 ^bB^	15.57 ± 0.58 ^aA^	15.87 ± 0.67 ^aA^
	14	12.52 ± 0.41 ^dB^	16.69 ± 0.48 ^aA^	15.27 ± 0.26 ^bA^	14.01 ± 0.62 ^cB^	12.01 ± 0.21 ^dB^
	28	14.52 ± 0.33 ^abA^	11.51 ± 0.41 ^cB^	12.14 ± 0.34 ^cB^	13.54 ± 0.35 ^bB^	15.28 ± 0.59 ^aA^
Chewiness	1	9.30 ± 0.08 ^cC^	14.54 ± 0.31 ^aA^	10.87 ± 0.40 ^bB^	14.00 ± 0.38 ^aA^	14.21 ± 0.61 ^aB^
	14	11.94 ± 0.31 ^bB^	14.62 ± 0.47 ^aA^	14.32 ± 0.27 ^aA^	15.08 ± 0.70 ^aA^	10.63 ± 0.41 ^cC^
	28	15.47 ± 0.22 ^aA^	10.03 ± 0.78 ^cB^	10.90 ± 0.89 ^bcB^	11.99 ± 0.64 ^bB^	15.70 ± 0.19 ^aA^
Syneresis (%)	1	20.52 ± 0.41 ^abB^	21.27 ± 2.18 ^aA^	18.99 ± 1.5 ^acAB^	17.71 ± 0.56 ^bcB^	16.60 ± 0.82 ^cB^
	14	21.31 ± 1.63 ^aA^	20.99 ± 1.32 ^aAB^	19.15 ± 0.98 ^abA^	19.54 ± 1.01 ^abA^	16.73 ± 0.23 ^bB^
	28	19.50 ± 0.66 ^caC^	20.66 ± 0.11 ^aB^	18.34 ± 0.09 ^cdB^	18.09 ± 0.61 ^cdB^	17.34 ± 0.15 ^dA^

Note: YC: control goat yogurt; YLE1%: goat yogurt with 1% addition of LE; YLE2%: goat yogurt with 2% addition of LE; YFE1%: goat yogurt with 1% addition of FE; YFE2%: goat yogurt with 2% addition of FE; ^a–d^ mean ± standard deviation with different lowercase letters on the same line differed by Tukey’s test (*p* < 0.05) between treatments; ^A–C^ mean ± standard deviation with different capital letters in the same column differed by Tukey’s test (*p* < 0.05) over storage time.

**Table 7 foods-13-03942-t007:** The sum of preference ranking for goat yogurts made with different concentrations (1 and 2%) of the flower extract (FE) and leaf extract (LE) of malvaviscus.

Ranking					
Formulations	YC	YLE1%	YLE2%	YFE1%	YFE2%
Sum of ranks	208 ^b^	320 ^c^	315 ^c^	189 ^b^	168 ^a^

Note: Lowercase letters within the same row indicate significant differences at the 5% level; F critical = 39. YC: control goat yogurt; YLE1%: goat yogurt with 1% addition of LE; YLE2%: goat yogurt with 2% addition of LE; YFE1%: goat yogurt with 1% addition of FE; YFE2%: goat yogurt with 2% addition of FE.

## Data Availability

The original contributions presented in the study are included in the article/Appendix A, further inquiries can be directed to the corresponding author.

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
