# Peer review of "Effect of Malvaviscus arboreus Flower and Leaf Extract on the Functional, Antioxidant, Rheological, Textural, and Sensory Properties of Goat Yogurt"

_foods, 2024, doi:10.3390/foods13233942_

Round 1

Reviewer 1 Report

Comments and Suggestions for Authors

Introduction

Line 41: replace renders with makes

Line 44: please rephrase

Line 52: replace possesses

Line 71: Please rephrase

Are there any studies on testing the use of Malvaviscus arboreus in food products? Please add.

2.1 Line 76-83: Please rephrase the materials. You should divide the materials: milk, malvaviscus, starter microorganism, probiotic cultures.

2.2 Please, describe the extraction procedures of the flowers and leaf since it is a key point for the manuscript. Did you use any chemical solvent? What was the humidity of the materials at the beginning? And what the final values? What was their appearance?

Figures of the yoghurt should be provided to see the differences in the appearance. 

2.3 Why did you choose 1% and 2%? Please, clarify.

How did you add the ingredients in the formulation? As a powder?

Line 217-225. Please explain better the results related to the decrease of acidity and pH of the yogurt. Is it possible a concomitant decrease?

Line 235-246. Explain better the results. Is it a positive effect the inhibition of starter cultures in such a product?

Line 252: how do the moisture affect these properties?

Line 256: did the inclusion of these ingredients change the yogurt matrix?

Line 273. Please explain better the change of the color in relationship with the oxidation of polyphenols and pH variation

3.1 The evaluation of polyphenolic content and antioxidant activity should be carried out to understand the starting value. 

This paragraph should be better written with a depth analysis of the results

3.4 Line 342 to 344 “…suggesting that the extracts acted as preservatives by inhibiting microbiological growth, including lactic acid strains” is it a positive effect?

Line 472. Reference 109 is not in the list of references. 

Comments on the Quality of English Language

Please check the English grammar all over the text and the captions of the tables

Author Response

Thank you for your thorough review. All the requested changes and additional information have been addressed and are included in the revised document attached. 

Reviewer 2 Report

Comments and Suggestions for Authors

The paper aims to investigate the effect of the addition of Malvaviscus arboreus flower and leaf extract on the functional, antioxidant, rheological, textural and sensory properties of goat yogurt.

The research is interesting and well conducted, and the paper is well written. The number of tests that the authors carry out are truly many and interesting, and provide a complete picture (physical and physico-chemical parameters and colour, bioactive compounds and total antioxidant activity, lipid oxidation, sanitary quality and viability of lactic acid bacteria, texture profile and syneresis, rheological behavior, sensory analysis) of the characteristics of the four different formulations: YFE1% (yogurt with 1% of flower extract); YFE2% (yogurt with 2% of flower extract); YLE1% (yogurt with 1% of leaf extract) and YLE2% (yogurt with 2% of leaf extract), compared with the control (normal yogurt without addictions).

Also from an English language point of view, the paper is well written, and fully smooth. Overall, the changes that should be made are all of a formal and non-substantial nature, so, for this reason, we recommend to the Editor a minor revision.

General comments:

The most important issue is this: you yourself claim that malvaviscus extracts have shown in the past a potent activity against microorganisms. In fact, also in your research work, you have recorded an inhibition of starter cultures by the extracts, as observed in the viability results of lactic acid bacteria. But you also claim that viability is not a determining factor for lactic bacteria to promote consumer health benefits. To a certain extent I can agree with you on this statement, but a yogurt is still made of lactic bacteria, which are its main characteristic, and the fact that they are inhibited could be a problem. For example, after 28 days what happens? Does the viability decrease further? Wouldn't it be a good idea to increase (even substantially) the initial inoculum of lactic acid bacteria?

A second important issue concerns the name of microorganisms. I am aware that the nomenclature of bacteria has recently been changed, but you use the new and the old one alternatively, and sometimes with the wrong name. For example:

Page 2 line 81: Streptococcus salivarius subsp. thermophillus: first of all, it is not thermophillus but thermophilus. Then, this is the old name. The new one, which has remained the same even after the latest changes, is Streptococcus thermophilus, as you have written elsewhere in the text. Be careful, also in Table 4 it is written “thermophillus” (and you should also add the “h” of “ther”).

Page 1 line 23, Page 4 line 165, page 9 line 345 (this time you must also put the name in italics), and Table 4: L. bulgaricus: the correct name, even before the new change in nomenclature, is L. delbrueckii subsp. bulgaricus, as you have written in other parts of the text.

In short, you should standardize the bacterial nomenclature, and above all write it correctly.

Here below there are the detailed comments:

Page 1 line 20: “FR” should be changed with “FE”.

Page 3 lines 115-116: you should write which type of methodology is: I mean, if it is a spectrophotometric IR method, it should be added in the text.

Page 3 line 142: be careful, it is the same title as in line 130, but I think it should be different, isn’t it?

Page 4 lines 184-185: some symbols are not in line with the text.

Page 5 line 215: You should write “Results and discussion”, because it is in effect a description of results with included a well-made discussion.

Page 5 lines 219-220: why also pH is reduced if titratable acidity diminishes? Shouldn't they be inversely proportional? I know that this is not always the case, because they essentially measure two different things. But, in this case, I would remove the sentence, because in any case the decrease in pH is not statistically significant, and differences that are not statistically significant should not be reported in the text.

Page 6 line 238: a comma should be put after “metabolism”.

Page 7 line 272: It would be interesting and more complete to also describe the other parameters present in Table 1 that gave statistically significant differences, especially between the five different formulations.

Page 8 line 327: You have included in the supplementary material some tables and figures that instead it would be interesting to show among the main ones. In particular, I would transfer among the main ones this Table S2, because are important results.

Page 11 lines 392-427: It is a very good discussion, compliments.

Page 12: Also Figure S2 and Table S9 should be transferred into the main ones, because are important results.

Page 12 line 480: in the conclusion it is better not to use the abbreviations, so “sample YFE2%” should be written as: “the treatment with the addition of 2% flower extract”.

Conclusions: in the conclusion, you should state what is, in your opinion, the best formulation among the four, and for which reasons.

Author Response

(The authors gave the same response as above.)
